# Evaluation of a Method for Converting SAGE Extinction Coefficients to Backscatter Coefficient for Intercomparison with LIDAR Observations

Travis N. Knepp[1], Larry Thomason[1], Marilee Roell[1], Robert Damadeo[1], Kevin Leavor[2],
Thierry Leblanc[3], Fernando Chouza[3], Sergey Khaykin[4], Sophie Godin-Beekmann[4], and David Flittner[1]

[1]NASA Langley Research Center, Hampton, Virginia 23681, USA
[2]Science Systems and Applications, Inc. Hampton, Virginia 23666, USA
[3]Jet Propulsion Laboratory, California Institute of Technology, Wrightwood, CA 92397
[4]LATMOS/IPSL, Sorbonne Universitè, UVSQ, CNRS, Paris, France

**Correspondence:** Travis N. Knepp (travis.n.knepp@nasa.gov)

**Abstract.** Aerosol backscatter coefficients were calculated using multi-wavelength aerosol extinction products from the SAGE II and III/ISS instruments. The conversion methodology is presented followed by an evaluation of the conversion algorithm's robustness. The SAGE-based backscatter products were compared to backscatter coefficients derived from ground-based lidar at three sites (Table Mountain Facility, Mauna Loa, and Observatoire de Haute-Provence). Further, the SAGE-derived lidar ratios were compared to values from previous balloon and theoretical studies. This evaluation includes the major eruption of Mt. Pinatubo in 1991 followed by the atmospherically quiescent period beginning in the late nineties. Recommendations are made regarding the use of this method for evaluation of aerosol extinction profiles collected using the occultation method.

## 1 Introduction

Stratospheric aerosol consists of sub-micron particles (Chagnon and Junge, 1961) that are composed primarily of sulfuric acid and water (Murphy et al., 1998) and play a crucial role in atmospheric chemistry and radiation transfer (Pitts and Thomason, 1993; Kremser et al., 2016; Wilka et al., 2018). Background stratospheric sulfuric acid is supplied by chronic, natural, emission of $CS_2$ (carbon disulfide), OCS (carbonyl sulfide), DMS (dimethyl sulfide), and $SO_2$ (sulfur dioxide), from both land and ocean sources (Kremser et al., 2016). The amount of sulfur in the stratosphere can be acutely, yet significantly, impacted by volcanic eruptions. This influence is not limited to relatively rare injections from large volcanic events such as the Mt. Pinatubo eruption of 1991 (McCormick et al., 1995), but episodic injections from smaller eruptions have been shown to have significant impact as well (Vernier et al., 2011). Therefore, ongoing long-term observations of stratospheric aerosol are important from both a climate and chemistry perspective.

The Stratospheric Aerosol and Gas Experiment (SAGE) is a series of satellite-borne instruments that use the occultation method (both solar and lunar light sourced) and have a lineage that spans four decades, originating with the Stratospheric Aerosol Measurement II in 1978 (SAM-II, Chu and McCormick (1979)). Using the occultation technique, the SAM II and SAGE instruments made direct measurements of vertical profiles of aerosol extinction coefficient ($k$, herein referred to simply as aerosol extinction) by recording light transmitted through the atmosphere from the sun/moon as it rises or sets. This attenuated light was then compared to exo-atmospheric values that were recorded when the light source was sufficiently high above the atmosphere. This technique allows for high-precision measurements on the order of 5%, as reported in the level-two data product, for SAGE aerosol extinction in the main aerosol layer. In general, stratospheric aerosol extinction measurements are challenging due to the paucity of aerosols under background conditions and the ephemeral nature of ash/particulates injected directly from volcanic eruptions. However, occultation observations have the benefit of long path lengths (on the order of 100-1000 km, dependent on altitude). Further, due to the self-calibrating nature of this method, SAGE measurements are inherently stable (i.e. minimal impact from instrument drift) and ideal for long-term trend studies.

Due to the SAGE instrument's level of precision and the limited aerosol number density in the stratosphere, validating the aerosol extinction products has proven challenging. Successful validation is further limited by the measured parameter itself since coincident stratospheric extinction measurements are scarce. Conversely, high-quality backscatter measurements from ground-based lidar instruments are more common and, despite operating at a fixed location, may provide sufficient coincident observations for an evaluation of the SAGE aerosol product. However, the backscatter and extinction coefficient products are not directly comparable.

Previous researchers have accomplished this comparison by application of conversion coefficients determined from balloon-borne optical particle counters (OPC, see Jäger and Hofmann (1991); Jäger et al. (1995); Jäger and Deshler (2002a)) or by selection of a wavelength-dependent lidar ratio ($S$, typically $\approx$40-46 sr; see Kar et al. (2019)) to invert the lidar backscatter (532 nm) to extinction, followed by wavelength correction to account for the differing SAGE/lidar wavelengths (conversion is carried out using the Ångström coefficient, Kar et al. (2019)). A major limitation of the balloon-based conversions is the uncertainty in the conversion factors (on the order of $\pm$30-40% Deshler et al. (2003); Kovilakam and Deshler (2015)) and the requirement for ongoing OPC launches to accurately observe both zonal and seasonal variability. The primary limitation of the lidar conversions is the challenge of appropriately selecting $S$. Indeed, Kar et al. (2019) showed that $S$ is both altitude and latitude dependent, and varies from 20–30 sr while other reports (Wandinger et al., 1995; Kent et al., 1998) have shown $S$ to go as high as 70 during background conditions. While a lidar ratio of 40-50 sr has been regarded as a satisfactory assumption, $S$ is, ultimately, uninformed about the atmosphere in which the measurement was recorded, making appropriate selection of $S$ difficult.

On the other hand, Thomason and Osborn (1992) invoked an eigenvector analysis, based on SAGE II extinction ratios, to convert extinction coefficients to total aerosol mass and backscatter coefficients to enable comparison with lidar observations. This method provided coefficients with uncertainties on the order of $\pm$20-30% and has been used in subsequent studies to convert lidar backscatter to extinction for comparison with SAGE observations (Osborn et al., 1998; Lu et al., 2000; Antuña

et al., 2002, 2003). As this method relied on SAGE-observed extinction coefficients it was more similar to our method than backscatter-to-extinction methods (vide supra) and may be considered a precursor to the present work.

Contrary to previous efforts to compare extinction and backscatter coefficients, the extinction-to-backscatter (EBC) method proposed in this study required relatively basic assumptions about the character of the underlying aerosol. These assumptions include composition, particle shape, and shape of the size distribution (common assumptions in Mie theory as further discussed below). While combining Mie theory and extinction measurements to gain insight into the nature of stratospheric aerosol is a common methodology (e.g. Hansen and Matsushima (1966); Heintzenberg, Jost (1982); Hitzenberger and Rizzi (1986);

Thomason (1992); Bingen et al. (2004)), the difference in our method is that we make no attempt to infer aerosol properties such as number density or particle size distribution. Instead, we apply Mie theory to infer the relationship between extinction and backscatter and use the range of the solution space of aerosol properties as a bounding box for uncertainty. Fortunately, within the regime of the available observations, this methodology is less sensitive to specific aerosol properties such that we can reasonably convert SAGE extinction to a derived backscatter for comparison with lidar. To this end, we present a

method of converting SAGE-observed extinction coefficients to backscatter coefficient for direct comparison with stratospheric lidar observations. This method is presented as an alternative evaluation technique for the SAGE products with the intent of expanding our long-term trend intercomparison opportunities (i.e. to include ground-based lidar as well as the possibility of satellite-borne lidar).

## 2    Instrumentation

### 70    2.1    SAGE II

The SAGE instruments used in the current study are the SAGE II (Oct. 1985 – Aug. 2005) and SAGE III on the International Space Station (SAGE III/ISS, June 2017 – Present, hereafter referred to as SAGE III) instruments. The SAGE II instrument and algorithm (v7.0) have been described previously by Mauldin et al. (1985) and Damadeo et al. (2013), respectively. The SAGE III instrument was described by Cisewski et al. (2014), and the algorithm (v5.1) will be the topic of upcoming publications. A

brief description will be offered here, but the reader is directed to these publications for details.

SAGE II was a seven-channel solar occultation instrument (386, 448, 452, 525, 600, 935, 1020 nm) that flew on the Earth Radiation Budget Satellite (ERBS) from October 1984 through August 2005. Due to the orbital inclination and the method of observation, SAGE II observations were limited to ≈30 occultations per day, with 3-4 times more observations at mid-latitudes than at tropical and high latitudes as seen in Fig. 1 panel A. The standard products included number density of gas-phase

species ($O_3$, $NO_2$, and $H_2O$) and aerosol extinction (385, 453, 525, and 1020 nm) with a vertical resolution of ≈1 km (reported every 0.5 km). The SAGE II v7.0 products were used in the current analysis.

SAGE III is a solar/lunar occultation instrument that is docked on the ISS and has a data record beginning in June 2017. The on-board spectrometer is a charge coupled device with resolution of 1-2 nm. The spectrometer's spectral range extends from 280 – 1040 nm in addition to a lone InGaAs photodiode at 1550 nm. Similar to SAGE II, SAGE III has a higher frequency

of observations at mid-latitudes as compared to the tropics and high latitudes (Fig. 1 panel B). The standard products include

number density of gas-phase species for both solar ($O_3$, $NO_2$, and $H_2O$) and lunar ($O_3$ and $NO_3$) observations, and aerosol extinction coefficient (384, 448, 520, 601, 676, 755, 869, 1020, 1543 nm), with 0.75 km resolution (reported every 0.5 km). The SAGE III v5.1 products (July 2017-September 2019) were used in the current analysis.

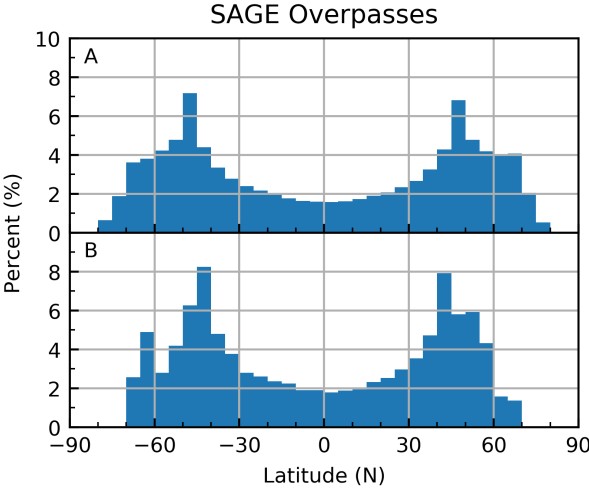

**Figure 1.** Zonal overpass bar charts for SAGE II (A) and SAGE III (B) missions.

## 2.2 Ground LIDAR

Ground lidar data from three stations were used within this study. To allow intercomparison with both SAGE II and SAGE III, candidate ground stations with a long-duration data record were preferred. Further, data quality is likewise important. The Network for Detection of Atmospheric Composition Change (NDACC, https://www.ndacc.org) was founded to observe long-term stratospheric trends by making long-term/high-quality atmospheric measurements. Therefore, stations within this network were selected for comparison. We identified three stations that satisfied the requirements of this analysis: Table Mountain Facility, Mauna Loa Observatory, and Observatoire de Haute-Provence. A brief description of these instruments and their algorithms is provided below.

### 2.2.1 Table Mountain Facility

The NASA Jet Propulsion Laboratory's (JPL) Table Mountain Facility (TMF) is located in southern California (34.4 N, 117.7 W, Alt. 2285 m). Backscatter coefficients derived from the ozone DIfferential Absorption Lidar (DIAL) were used in the current study and have a record extending back to the beginning of 1989 (McDermid et al., 1990b, a). The lidar used the third harmonic of a Nd:YAG laser to record elastic backscatter at 355 nm, which was corrected for ozone and $NO_2$ absorption, and Rayleigh extinction. The corrected backscatter was then used to calculate the aerosol backscatter coefficient from backscatter ratio (BSR) (Northam et al., 1974; Gross et al., 1995). Prior to June 2001 the BSR was calculated using pressure and temperature data from a National Centers for Environmental Protection (NCEP) meteorological model. Since June 2001 the BSR has been computed

using the 387 nm channel from a newly installed Raman channel as the purely-molecular component in the BSR. For both cases, the BSR was normalized to 1 between 30-35 km where it was assumed that the aerosol backscatter contribution was negligible.

### 2.2.2 Mauna Loa Observatory

The NOAA Mauna Loa Observatory (MLO, 19.5 N, 155.6 W, Alt. 3.4 km) is located on the Big Island of Hawai'i. The dataset used here comes from the elastic and inelastic (Raman) backscatter channels of the JPL ozone DIAL that began measurements in 1993 (McDermid et al., 1995; McDermid, I. Stuart, 1995). Just like the JPL-TMF system, the lidar used the third harmonic of a Nd:YAG laser to record the elastic backscatter at 355 nm, followed by correction for ozone and $NO_2$ absorption, and Rayleigh extinction. The corrected backscatter was then used to calculate the aerosol backscatter coefficient from the backscatter ratio using the 387 nm channel as the purely-molecular component in the BSR as described in Chouza et al. (2020). The BSR was normalized to 1 at a constant altitude of 35 km where it was assumed that the aerosol backscatter contribution was negligible.

### 2.2.3 Observatoire de Haute-Provence

The Observatoire de Haute-Provence (OHP, 43.9 N, 5.7 E, 670 m ASL) is located in Southern France and has an elastic backscatter lidar record that began in 1994 and is based on DIAL ozone measurements that began in 1985. In 1993, the lidar system was updated for improved measurements in the lower stratosphere (Godin-Beekmann et al., 2003; Khaykin et al., 2017). The lidar used the third harmonic of an Nd:YAG laser (355 nm) to record elastic backscatter, followed by inversion using the Fernald-Klett method (Fernald, 1984; Klett, 1985) to provide backscatter and extinction coefficients, assuming an aerosol-free region between 30-33 km and a constant lidar ratio of 50 sr. The error estimate for this method is <10% (Khaykin et al., 2017).

## 3 Methodology

Extinction and backscatter observations cannot be directly compared. In order to evaluate the agreement between backscatter measurements and extinction coefficient measurements, the data types must be converted to a common parameter, thereby requiring a conversion algorithm. As previously mentioned, this is usually done by converting backscatter to extinction coefficient using conversion factors from sources independent of either instrument (e.g. constant lidar ratio). Herein, we derive a process to infer this relationship based on the spectral dependence of SAGE II/III aerosol extinction coefficient measurements and only make basic assumptions on the character of the underlying aerosol. Indeed, this EBC method is proposed to act as a bridge between aerosol extinction and backscatter observations. This bridge is founded upon Mie theory (Kerker, 1969; Hansen and Travis, 1974; Bohren and Huffman, 1983) and invokes the typical assumptions required in Mie-theory models: particle shape, composition, and distribution shape and width. Herein we assumed all particles are spherical, are composed primarily of sulfate (75% $H_2SO_4$, 25% $H_2O$ by mass (Murphy et al., 1998)), and that the particle size distribution (PSD) is single-mode log-normal. Refractive index values from Palmer and Williams (1975) were used in the calculations.

Particulate backscatter and extinction efficiency factors ($\boldsymbol{Q}_{sca}(\lambda, \boldsymbol{r})$ and $\boldsymbol{Q}_{ext}(\lambda, \boldsymbol{r})$, respectively; for derivation of $\boldsymbol{Q}(\lambda, \boldsymbol{r})$
see Kerker (1969) and Bohren and Huffman (1983)) were calculated for a series of particle radii ($\boldsymbol{r} = [1, 2, \ldots, 1500]$ nm)
and incident light wavelengths ($\lambda = [350, 351, \ldots, 2000]$ nm). Subsequently, a series of log-normal distributions ($\boldsymbol{P}(r_m, \sigma_g)$,
described by Eq. 1 where $\sigma_g$ is the geometric standard deviation and $r_m$ is the mode radius (the median radius of a log-normal
distribution is commonly referred to as mode radius in aerosol literature; we adopt this convention here)) were calculated for
the same family of particles with five distribution widths ($\sigma_g = [1.2, 1.4, 1.5, 1.6, 1.8]$) that were chosen to cover the range
of likely distribution widths (Jäger and Hofmann, 1991; Pueschel et al., 1994; Fussen et al., 2001; Deshler et al., 2003).
This was performed for all 1500 radii to calculate a new log-normal distribution as $r_m$ took on each value within $\boldsymbol{r}$. Values for
$\boldsymbol{Q}_{sca}(\lambda, \boldsymbol{r})$, $\boldsymbol{Q}_{ext}(\lambda, \boldsymbol{r})$, and $\boldsymbol{P}(r_m, \sigma_g)$ were then fed into Eqs. 2 & 3 to produce three-dimensional lookup tables ($r_m \times \lambda \times \sigma_g$)
of extinction and backscatter ($\boldsymbol{k}(\lambda, r_m, \sigma_g)$ and $\boldsymbol{\beta}(\lambda, r_m, \sigma_g)$, respectively; hereafter referred to as $k$ and $\beta$) coefficients as a
function of mode radius, incident light wavelength, and distribution width.

At this point a technical note regarding construction of the log-normal distribution must be made. Construction of a log-
normal distribution fails when the mode radius is near the limits of $r_m$ (i.e. 1 or 1500 nm), yielding a truncated log-normal
distribution. However, the mode radii required for this analysis (i.e. to generate the corresponding SAGE extinction ratios)
ranged from $\approx$50 to $\approx$500 nm, well away from these bounds. With the backscatter and extinction lookup tables thus created,
we now focus on their utilization in converting from $k$ to $\beta$.

$$\boldsymbol{P}(r_m, \sigma_g) = \frac{1}{\sqrt{2\pi}\ln(\sigma_g)\boldsymbol{r}} \exp\left[\frac{(\ln(\boldsymbol{r}) - \ln(r_m))^2}{-2\ln(\sigma_g)^2}\right] \tag{1}$$

$$\boldsymbol{k}(\lambda, r_m, \sigma_g) = \int \pi \boldsymbol{r}^2 \boldsymbol{P}(r_m, \sigma_g) \boldsymbol{Q}_{ext}(\lambda, \boldsymbol{r}) dr \tag{2}$$

$$\boldsymbol{\beta}(\lambda, r_m, \sigma_g) = \frac{1}{4\pi} \int \pi \boldsymbol{r}^2 \boldsymbol{P}(r_m, \sigma_g) \boldsymbol{Q}_{sca}(\lambda, \boldsymbol{r}) dr \tag{3}$$

Wavelengths were selected based on SAGE extinction channels and available lidar wavelength, and the lookup tables were
used to create the plots in Fig 2. Though this figure only shows data for one combination of extinction and backscatter wave-
lengths, similar figures were generated for each combination (not shown) with the 520/1020 combination providing the best
combination of linearity, atmospheric penetration depth, and wavelength overlap between SAGE II and SAGE III. This figure
elucidates the relationship between the inverted lidar ratio ($\beta/k$, hereafter referred to as $S^{-1}$), extinction ratio, and distribu-
tion width. Indeed, this figure provided the nexus between extinction and backscatter observations and between theory and
observation since SAGE-observed extinctions were imported into this model to derive $\beta_{355}$. To do this, SAGE extinction ratios
($k_{520}/k_{1020}$) were used to define the abscissa value, followed by identifying the ordinate value ($S^{-1}$) according the line drawn
in Fig. 2 followed by multiplication by the SAGE-observed $k_{1020}$. For example, if the observed SAGE extinction ratio was
6, then $S^{-1} \approx$0.2 when $\sigma_g$=1.6, and the SAGE-derived backscatter coefficient ($\beta_{SAGE}$) can be calculated via Eq. (4), where

$k_{1020}$ is the SAGE extinction product at 1020 nm. It is important to note a departure from convention in how the $S^{-1}$ values are reported in Fig. 2. The standard convention would require both coefficients to be at the same wavelength. The current methodology requires these coefficients to be at different wavelengths as explained above. This deviation is only made in this conversion step (i.e. when using the data presented in Fig. 2), while subsequent discussion of lidar ratio estimates (e.g. Tables 2, 3, Figs. 6, 9, and §4.1.2) use the conventional lidar ratio definition.

$$\beta_{SAGE} = S^{-1} \cdot k_{1020} \tag{4}$$

A potential limitation of this method is that, for large particle sizes (extinction ratios < 1 in Fig. 2, corresponding to mode radius of $\approx$500 nm), two solutions for $S^{-1}$ are possible. Further, for smaller particles sizes (extinction ratios > 6 in Fig. 2, corresponding to mode radius $\approx$50) the solutions rapidly diverge as a function of $\sigma_g$, making selection of $\sigma_g$ increasingly important. However, SAGE extinction ratios were rarely outside these limits. This is seen in Fig. 3 where probability density functions (PDFs) and cumulative distribution functions (CDFs) of stratospheric extinction ratios were plotted for SAGE II and SAGE III. The stark difference in distribution shape between panels A and B is due to the SAGE II mission being dominated by volcanic eruptions, while the SAGE III mission, to date, has experienced a relatively quiescent atmosphere. Data in panels A and C of Fig. 3 were broken into two periods: 1. where the atmosphere was impacted by the Mt. Pinatubo eruption (1-June 1991 – 1-January 1998), 2. periods where the impact of Pinatubo was expected to be less significant. It was observed that the majority of extinction ratios (>90%) were between one and six regardless of Pinatubo's impact. Therefore, we conclude that the majority of SAGE's observations can take advantage of this methodology.

While Fig. 3 shows that most extinction ratios avoid either multiple solutions or significant divergence in solutions due to $\sigma_g$, it is understood that, due to uncertainty in $\sigma_g$ there is an associated uncertainty in the derived $\beta_{355}$. To account for this spread, SAGE-based backscatter coefficients were calculated for both extremes of $\sigma_g$ (i.e. 1.2 and 1.8). These two solutions were plotted in subsequent figures to illustrate this spread. Further discussion of uncertainties associated with the selection of $\sigma_g$ is presented in §3.2.

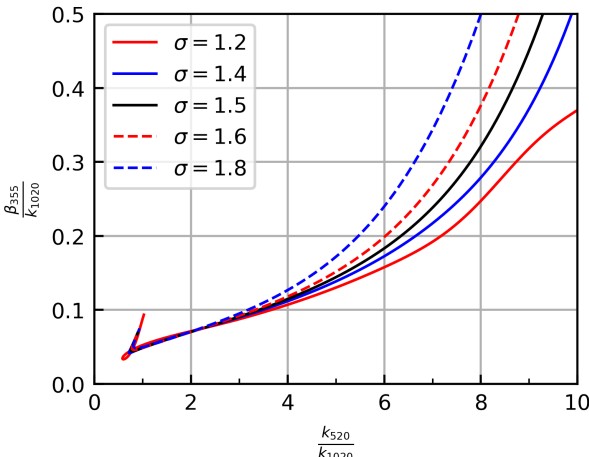

**Figure 2.** Theoretical relationship between the inverted lidar ratio ($S^{-1}$) and extinction ratio. Wavelength selection was based on the ground-based backscatter lidars and available SAGE channels.

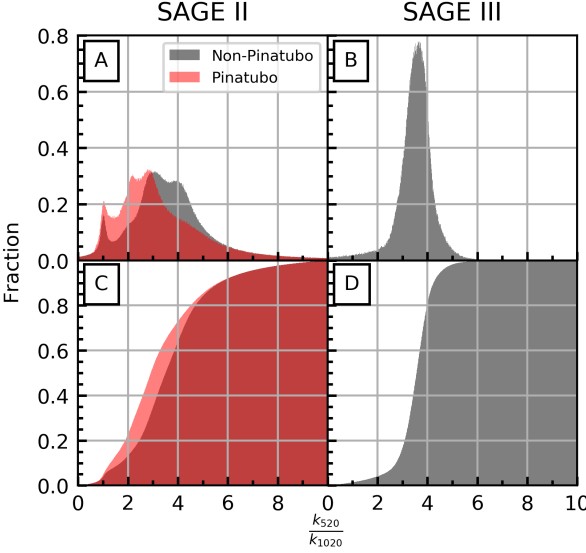

**Figure 3.** Extinction ratio PDFs (panels A and B, bin width is 0.01) and CDFs (panels C and D) for the SAGE II and SAGE III missions. Due to the impact of the Pinatubo eruption, separate histograms were created for the SAGE II panels. Only stratospheric altitude (tropopause +2 km) were used to generate this figure to ensure exclusion of cloud contamination. In the SAGE II panels (A & C), light red indicates data impacted by the Pinatubo eruption, gray indicates data not impacted by Pinatubo, and dark red is the overlapped region of the two.

## 3.1 Internal Evaluation of the Method

Figure 2 showed the relationship between extinction ratio and $S^{-1}$ for one combination of wavelengths. Since SAGE II and SAGE III recorded extinction coefficients at multiple wavelengths, there were multiple wavelength combinations from which to choose. Under ideal conditions, the $\beta$ derived using this conversion methodology should be independent of wavelength combination. Indeed, it can be trivially demonstrated that, working strictly within the confines of theory (i.e. no noise or uncertainty), this is the case. However, in reality, the SAGE extinction products were impacted by errors originating in hardware (e.g. instrument noise), retrieval algorithm (e.g. how well gas species were cleared prior to retrieving aerosol extinction) and atmospheric conditions (e.g. impact of clouds). Therefore, the method's consistency was evaluated by calculating $\beta_{SAGE}$ using three wavelength combinations to form the abscissa of Fig. 2: 385/1020, 450/1020, and 520/1020 (hereafter this calculated $\beta$ is referred to as $\beta_{S(385)}$, $\beta_{S(450)}$, and $\beta_{S(520)}$). The target backscatter wavelength was held constant (355 nm) in this evaluation for two reasons: 1. this is the lidar wavelength used at the three ground sites used in this study, 2. selection of lidar wavelength does not influence the evaluation of the method's consistency.

### 3.1.1 Comparison of $\beta$

To evaluate the robustness of the EBC algorithm, $\beta_{SAGE}$ was calculated at three wavelength combinations: $\beta_{S(385)}$, $\beta_{S(450)}$, and $\beta_{S(520)}$ with the 520 ratio acting as the reference (i.e. the 385 and 450 nm ratios were compared to the 520 ratio in subsequent statistical analyses). The intent of this comparison was to quantify and qualify the variability between the differing $\beta_{SAGE}$ products. The following analysis was conducted using zonal statistics (5° latitude, 2 km altitude bins) that were weighted by the inverse measurement error within the reported SAGE extinction products. These data are presented both graphically (Figs. 4 and 5) and numerically (Table 1).

The zonal weighted coefficient of correlation ($R^2$) and weighted slope of linear regression profiles are presented in panels A-D of Figs. 4 and 5 for SAGE II and SAGE III, respectively. It was observed that the coefficients of correlation and slopes between the three products were high throughout the profile ($R^2 \geq 0.85$ and slope $\geq 0.78$; Table 1) and were higher towards the middle of the stratosphere ($R^2 > 0.95$ and slope $\approx 1$). However, at lower and higher altitudes the overall performance was worse. This degradation was driven by several factors: 1. the shorter wavelengths were attenuated higher in the atmosphere due to increasing optical thickness, which led to negligible transmittance through lower-sections of the atmosphere, 2. impact of cloud contamination at lower altitudes, 3. differences in the higher altitudes were the product of limited aerosol number density (i.e. increased uncertainty due to decreased extinction). To better understand this altitude dependence and identify altitudes where the conversion method may be most successfully applied, we evaluated a series of altitude-based filtering criteria. A brief discussion of these criteria, and their impact on the statistics in Table 1 will be presented prior to continued discussion of Figs. 4 and 5.

Correlation plots (not shown) were generated for each latitude band and each altitude from 12–34 km (2 km wide bins centered every 2 km) with corresponding regression statistics to better understand how the agreement between the backscatter products varied with altitude and latitude and to aid in defining reasonable filtering criteria to mitigate the impact of spurious

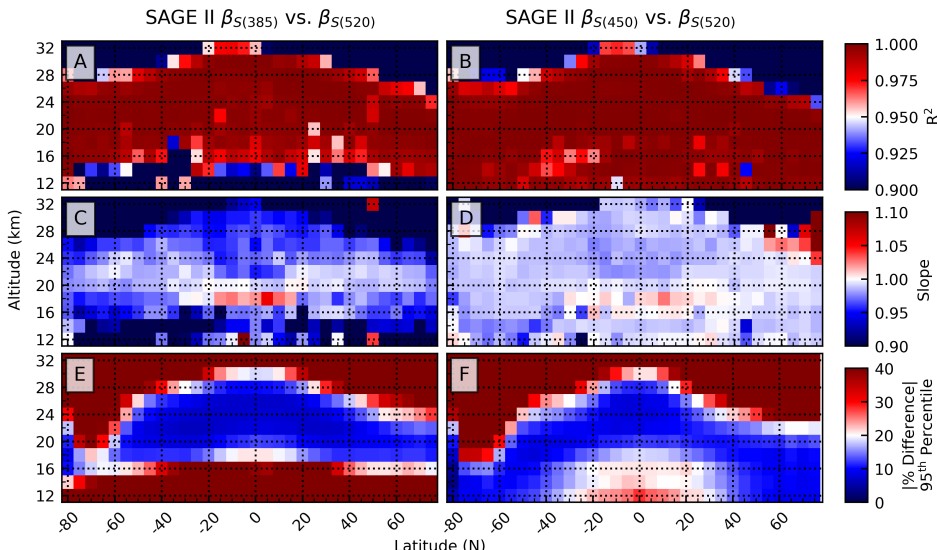

**Figure 4.** Zonal statistics displaying the overall agreement between the two backscatter calculations using data collected during the SAGE II mission. Panels E and F show the 95th percentile of the absolute value for the percent difference.

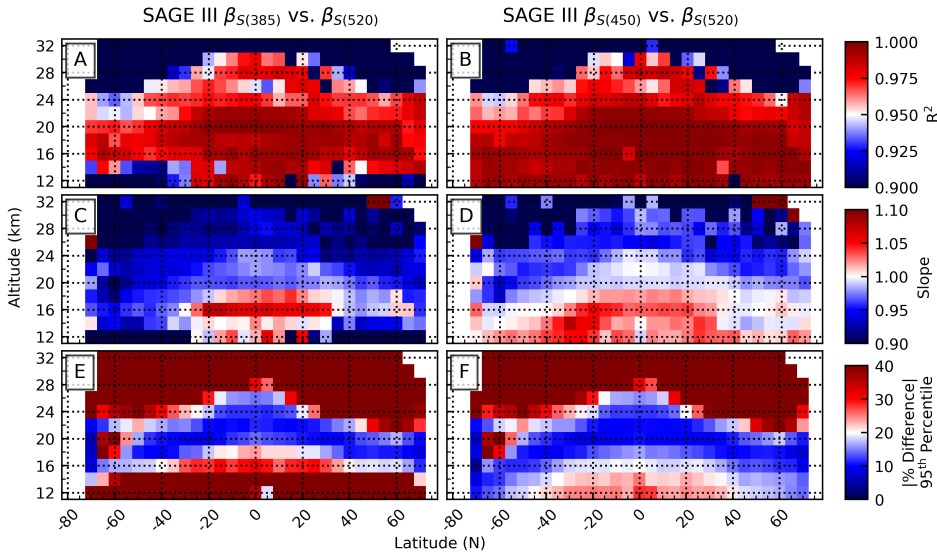

**Figure 5.** Same as in Fig. 4, but for the SAGE III mission.

retrieval products typically seen at lower and higher altitudes. We observed that data collected between 15 and 31 km had higher coefficients of correlation, slopes closer to one, and had a tighter grouping about the 1:1 line (i.e. fewer outliers in either

|  | SAGE II | | SAGE III | | SAGE-II & SAGE III |
|  | $\beta_{S(385)}$ | $\beta_{S(450)}$ | $\beta_{S(385)}$ | $\beta_{S(450)}$ | $\beta_{S(385)}$ & $\beta_{S(450)}$ |
|---|---|---|---|---|---|
| Slope Min | -0.08 (0.01) | -0.04 (0.58) | -0.19 (0.75) | -0.02 (0.54) | -0.19 (0.01) |
| Slope Max | 1.30 (1.06) | 1.25 (0.1.25) | 2.21 (1.19) | 34.68 (1.35) | 34.68 (1.35) |
| Slope Mean | 0.78 (0.94) | 0.89 (0.99) | 0.88 (0.95) | 1.02 (0.97) | 0.89 (0.96) |
| Slope Median | 0.95 (0.97) | 0.99 (0.99) | 0.94 (0.95) | 0.97 (0.97) | 0.97 (0.98) |
| Slope Standard Deviation | 0.32 (0.12) | 0.26 (0.05) | 0.23 (0.05) | 1.85 (0.06) | 0.92 (0.08) |
| Slope $P_{95}$ (all altitudes) | 0.01 – 1.30 | 0.15 – 1.25 | 0.24 – 1.52 | 0.37 – 1.67 | 0.12 – 1.67 |
| Slope $P_{95}$ (15-31 km) | 0.71 – 1.06 | 0.95 – 1.03 | 0.84 – 1.07 | 0.89 – 1.05 | 0.87 – 1.06 |
| $R^2$ Min | 0.28 (0.37) | 0.02 (0.58) | 0.00 (0.41) | 0.00 (0.32) | 0.00 (0.32) |
| $R^2$ Max | 1.00 (1.00) | 1.00 (1.00) | 1.00 (1.00) | 1.00 (1.00) | 1.00 (1.00) |
| $R^2$ Mean | 0.87 (0.98) | 0.90 (0.98) | 0.85 (0.95) | 0.87 (0.95) | 0.87 (0.97) |
| $R^2$ Median | 0.99 (1.00) | 1.00 (1.00) | 0.96 (0.97) | 0.98 (0.98) | 0.98 (0.99) |
| $R^2$ Standard Deviation | 0.20 (0.08) | 0.18 (0.07) | 0.24 (0.09) | 0.24 (0.10) | 0.22 (0.08) |
| $R^2$ $P_{95}$ | 0.41 (0.89) | 0.18 (0.89) | 0.17 (0.77) | 0.19 (0.78) | 0.34 (0.82) |

**Table 1.** Aggregate statistics for line-of-best-fit slope and $R^2$ for the SAGE II and SAGE III products as compared to $\beta_{S(520)}$. Values in parentheses were calculated after restricting the dataset to altitudes between 15–31 km. The last column was calculated using data from both missions and backscatter values calculated using both $\beta_{S(385)}$ and $\beta_{S(385)}$. The slope $P_{95}$ data show the range of slopes (centered on the mean).

direction). From this, we defined the altitude-based filtering criteria to only include data collected within the altitude range 15-31 km.

As an evaluation of how much influence data outside the 15–31 km range had on this analysis an ordinary line of best fit was calculated for each combination of beta values (i.e. $\beta_{S(385)}$ vs $\beta_{S(520)}$ and $\beta_{S(450)}$ vs $\beta_{S520}$) for the SAGE II and

225 SAGE III missions under two conditions: 1. all available data was used, 2. only data from 15-31 km was used. A summary of this evaluation is presented in Table 1 wherein it is observed that when all data throughout the profile was used the mean slope (0.78-1.02) and mean $R^2$ (0.87-0.98) had broad ranges, as did the corresponding standard deviations. However, when the dataset was limited to 15-31 km (values in parentheses) the range of mean slopes (0.94-0.99) and mean $R^2$ (0.95-0.98) decreased significantly, as did the corresponding standard deviations. It was observed that when the filtering criteria were in

230 place the standard deviation significantly narrowed, in some cases by more than an order of magnitude.

By considering only the mean slope and mean $R^2$ values the impact of the filtering criteria is partially masked. The influence of this criteria is better observed by considering the min/max values for both slope and $R^2$ and by considering its impact on the 95th percentile ($P_{95}$). Here, $P_{95}$ was calculated using a non-traditional method. For slope, $P_{95}$ represents the range over which 95% of the data fall, centered on the mean. As an example, if the mean slope is 1, how far out from 1 must we go before 95% of

235 the data are captured? This range is not necessarily symmetrical about the mean since either the minimum or maximum slope

may be encountered prior to reaching the 95% level. On the other hand, $P_{95}$ for $R^2$ indicates the lowest $R^2$ value required to capture 95% of the data ($R^2 = 1$ acting as the upper bound). Indeed, contrasting the full- and filtered-profile $P_{95}$ values in Table 1 provides a convincing illustration of the improvement the filtering criteria had on the comparison.

From this evaluation we conclude that data outside 15-31 km significantly influenced the statistics and that the applicability of this conversion method is limited to regions where sufficient signal is received by the SAGE instruments, namely 15-31 km.

Having established an altitude range interval over which the EBC method remains robust we can continue the evaluation of the aggregate statistics as shown in Figs. 4 and 5. To gauge the overall difference between the products, $P_{95}$ for the absolute percent differences are shown in panels E and F. This is an illustrative statistic in that it shows, for example, 95% of the time the $\beta_{S(450)}$ and $\beta_{S(520)}$ products for SAGE II were within 10% of each other at 24 km over the equator. More generally, it is observed that the two products were within $\pm20\%$ of each other (all wavelengths for both SAGE II and SAGE III) throughout most of the atmosphere, similar to the $R^2$ and slope products. Similar to the panels A-D, the absolute percent difference has better agreement between the longer wavelength products (within $\pm20\%$ for $\beta_{S(450)}$ and $\beta_{S(520)}$), and follows a similar contour to that seen in the $R^2$ (panels A and B) and slopes (panels C and D).

The high $R^2$ values and slopes are encouraging and we conclude that, throughout most of the lower stratosphere, the calculated backscatter coefficient is independent of SAGE extinction channel selection. It is noted that the performance of $\beta_{S(385)}$ was limited by comparatively rapid attenuation higher in the atmosphere, thereby limiting applicability of this channel within the EBC algorithm. Further, we suggest that this attenuation was the driving factor in the worse agreement between $\beta_{S(385)}$ and $\beta_{S(520)}$. Conversely, $\beta_{S(450)}$ showed better agreement with $\beta_{S(520)}$ throughout most of the lower stratosphere, leaving two viable extinction ratios for calculating $\beta_{SAGE}$: 450/1020 and 520/1020 nm. While the 450 nm channel will not be attenuated as high in the atmosphere as the 385 nm channel, it will saturate before the 520 nm channel.

In addition to comparing $\beta$ as a function of extinction wavelength, the algorithm performance can be compared, qualitatively, between the SAGE II and SAGE III missions. While this comparison is valid, it must be remembered that the SAGE II record extends over a twenty-year period, including impacts from the El Chichón (1982) and Pinatubo (1991) eruptions, which significantly influenced atmospheric composition. Conversely, the SAGE III mission is currently in its third year and, to date, has had no opportunity to observe the impact of a major volcanic eruption. On the contrary, current stratospheric conditions have been relatively clean for the past twenty years. The agreement in performance between the two missions is most readily seen by comparing the filtered slope and $R^2$ statistics in Table 1, wherein it is observed that the differences are statistically insignificant.

From this evaluation we determined that the selection of extinction wavelength combination had minimal impact on the calculated backscatter products when altitudes are limited to 15-31 km (i.e. each combination of SAGE wavelengths yielded the same backscatter coefficient within the provided errors). Therefore, we proceed with the current analysis by using the 520/1020 nm combination to convert SAGE-observed extinction coefficients to backscatter coefficient for comparison with lidar-observed backscatter coefficients.

## 3.2 Uncertainties

As with any study that involves modeling PSDs, the dominant sources of uncertainty are in the assumptions of aerosol composition and distribution parameters. Here, the particle number density and mode radius play a minor role. However, as seen in Fig. 2, selection of $\sigma_g$ has a variable impact. The statistics presented in §3.1.1 were calculated using $\sigma_g$=1.5, but are not influenced by the selection of $\sigma_g$ since changing the selection of $\sigma_g$ will shift all data sets up or down equally. On the other hand, the accuracy of the method is highly dependent on $\sigma_g$. As an example, setting $\sigma_g$=1.5 leads to a +32/-16% uncertainty (compared to $\sigma$=1.8 and $\sigma$=1.2, respectively) when the extinction ratio equals six. Since >90% of the stratospheric extinction ratios do not exceed six, we consider +32/-16% acts as a reasonable upper limit of expected uncertainty for this analysis. This uncertainty is depicted in subsequent figures by a shaded region that represents the extinction-ratio-dependent upper/lower bounds for $\beta_{S(520,\ \sigma_g=1.2)}$ and $\beta_{S(520,\ \sigma_g=1.8)}$ (i.e. for smaller extinction ratios the spread between $\beta_{S(520,\ \sigma_g=1.2)}$ and $\beta_{S(520,\ \sigma_g=1.8)}$ decreased). It was observed that this spread was negligible at lower altitudes, but increased with altitude.

Another challenge in comparing SAGE and lidar observations is the differing viewing geometries. The uncertainty introduced by these differing geometries cannot be easily accounted for. However, current versions of the algorithm (Damadeo et al., 2013) as well as previous studies (Ackerman et al., 1989; Cunnold et al., 1989; Oberbeck et al., 1989; Wang et al., 1989; Antuña et al., 2002; Jäger and Deshler, 2002b; Deshler et al., 2003) have taken advantage of the horizontal homogeneity of stratospheric aerosol, which mitigates the impact of differing viewing geometries.

## 4 Method Application

The EBC method was applied to SAGE II and SAGE III data sets for intercomparison with ground-based lidar products. A discussion of the results of each SAGE mission follows.

### 4.1 SAGE II

The SAGE II record spanned over 20 years and had the benefit of observing the impact of two of the largest volcanic eruptions of the twentieth century: recovery from El Chichón in 1982 and the full life cycle of the Mount Pinatubo eruption of 1991 followed by return to quiescent conditions in the late 1990s. Within this record the extinction and backscatter coefficients spanned nearly two orders of magnitude, providing an interesting case study.

SAGE II data were used to estimate $\beta_{355}$ using the 520/1020 extinction ratio (Fig. 2). For this comparison, $\beta_{SAGE}$ was calculated on a profile-by-profile basis. These profiles were then used to calculate zonal monthly means. Likewise, lidar profiles were averaged on a monthly basis for comparison. The time series, at four altitudes, are presented in Figs. 6, 7 and 8 for Table Mountain, Mauna Loa, and OHP respectively. The spread in the $\beta_{SAGE}$ value, due to varying results in solving Eq. (4) for differing values of $\sigma_g$, is represented by the black shaded time series data. It is noted that, most of the time, this shaded area is indistinguishable from the black line thickness. Error bars in Figs. 6, 7 and 8 represent the standard error (error on mean). We

observed that the data sets were in qualitatively good agreement at all altitudes, especially when the atmosphere was impacted by the Pinatubo eruption (June 1991 - 1998).

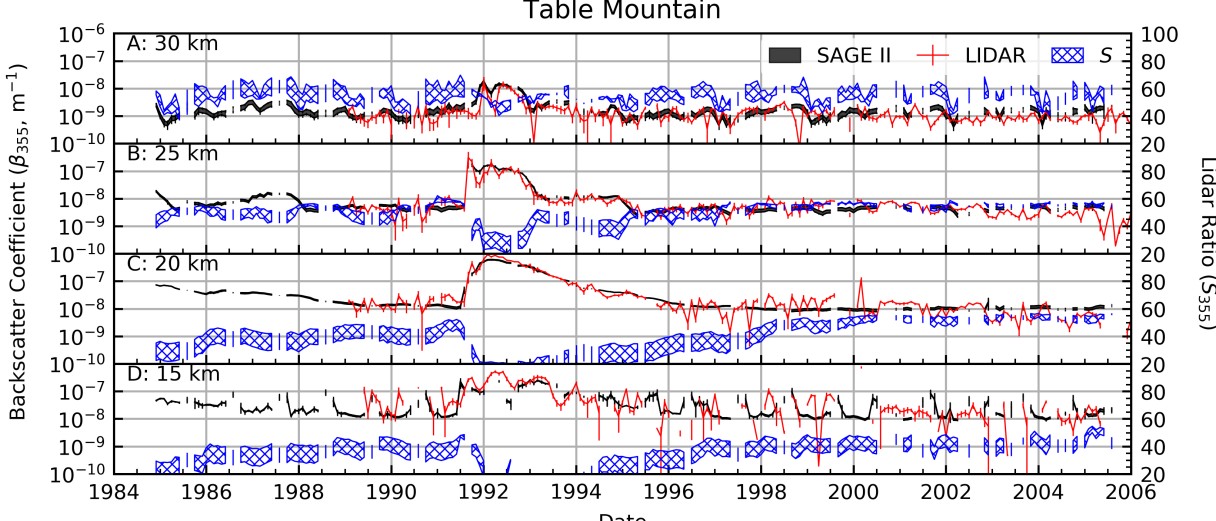

**Figure 6.** Time series of SAGE II (monthly zonal mean), lidar (monthly mean) backscatter coefficients and a SAGE-based lidar ratio estimate at 355 nm (monthly zonal mean) over the Table Mountain Facility. The spread in the SAGE-derived backscatter and lidar ratios (both coefficients at same wavelength) represent the range of values due to changing the spread ($\sigma$) of the log-normal distribution. The backscatter upper bound is when $\sigma=1.8$ and the lower bound is for $\sigma=1.2$ (vice versa for the lidar ratio). Error bars represent standard errors. Altitude bins span $\pm 2.5$ km.

Statistics for the time series data are presented in Table 2 below. The data were broken into two time periods: 1. when the signal was perturbed by the Pinatubo eruption (labeled PE in the table, June 1991–December 1997) as defined by Deshler et al. (2003, 2006)), 2. periods outside the Pinatubo impact, classified as background (labeled BG in the table). As seen in Figs. 6, 7, and 8, the return to background conditions was sooner at higher altitudes, which may influence some statistics in Table 2 since the Pinatubo time period classification (i.e. June 1991–December 1997) were applied to all altitudes. Statistics in this table were calculated using SAGE monthly zonal means and lidar monthly mean values at four altitudes. Percent differences were calculated using Eq. (5).

$$\%\text{Diff} = 100 * \frac{\beta_{SAGE} - \beta_{Lidar}}{0.5\left(\beta_{SAGE} + \beta_{Lidar}\right)} \tag{5}$$

### 4.1.1 TMO

Data collected at 15 km showed the worst agreement due to atmospheric opacity and cloud contamination as discussed above. Conversely, the agreement was best at 20 & 25 km (percent difference within $\approx$10%), where the atmosphere was most impacted

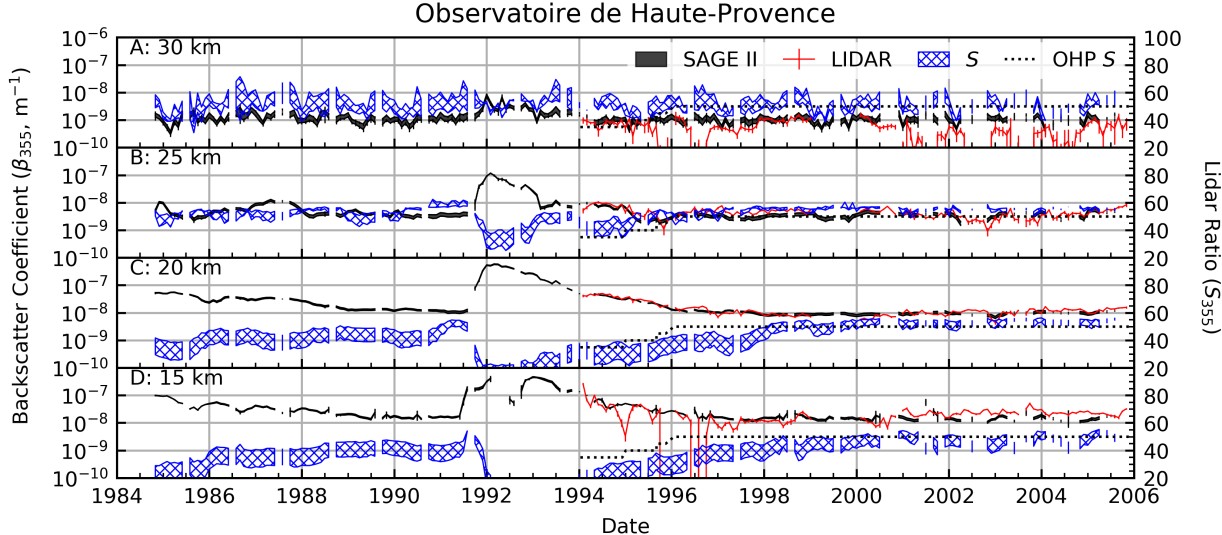

**Figure 7.** Same as Fig. 6, but for OHP. Dots mark the lidar ratio as estimated within the OHP record.

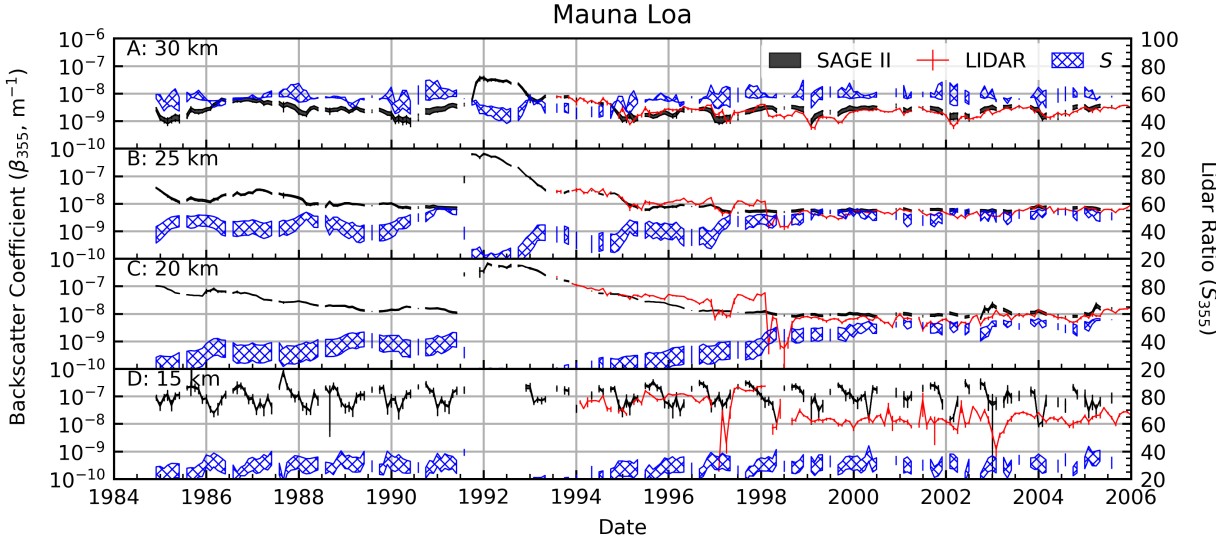

**Figure 8.** Same as Fig. 6, but for Mauna Loa.

by the Pinatubo eruption ($k_{520}$ increased by $\approx 2$ orders of magnitude) followed by an extended, approximately exponential, return to background conditions in the late 90s (Deshler et al., 2003, 2006). The stability of the lidar signal decreased as the amount of aerosol in the atmosphere decreased, beginning in $\approx 1996$ with more significant fluctuations appearing immediately

| Site | Parameter | Altitude (km) | $R^2_{PE}$ | $R^2_{BG}$ | Slope$_{PE}$ | Slope$_{BG}$ | %Diff$_{PE}$ | %Diff$_{BG}$ |
|------|-----------|---------------|------------|------------|--------------|--------------|--------------|--------------|
| TMO | $\beta$ | 30 | 0.72 | 0.00 | 0.70 | -0.04 | 20.84 (0.9) | 22.26 (0.7) |
| | $\beta$ | 25 | 0.82 | 0.01 | 0.87 | -0.36 | 11.35 (0.6) | 1.33 (0.7) |
| | $\beta$ | 20 | 0.94 | 0.00 | 1.35 | 0.13 | 6.01 (0.8) | -4.12 (0.8) |
| | $\beta$ | 15 | 0.56 | 0.00 | 1.28 | -0.38 | 5.87 (1.8) | -20.82 (1.7) |
| OHP | $\beta$ | 30 | 0.03 | 0.00 | -0.28 | 0.02 | 60.48 (1.8) | 86.80 (1.8) |
| | $\beta$ | 25 | 0.64 | 0.25 | 0.83 | 0.91 | -29.44 (0.7) | -4.40 (0.6) |
| | $\beta$ | 20 | 0.94 | 0.35 | 1.07 | 1.52 | -5.82 (0.4) | -1.88 (0.4) |
| | $\beta$ | 15 | 0.82 | 0.00 | 0.91 | -0.01 | -29.02 (0.2) | -21.54 (0.8) |
| OHP | $S$ | 30 | — | — | — | — | 16.52 (0.4) | 5.82 (0.2) |
| | $S$ | 25 | — | — | — | — | 16.08 (0.2) | 14.42 (0.1) |
| | $S$ | 20 | — | — | — | — | -16.97 (0.2) | 8.21 (0.1) |
| | $S$ | 15 | — | — | — | — | 17.32 (3.7) | -3.80 (0.1) |
| MLO | $\beta$ | 30 | 0.81 | 0.53 | 0.89 | 0.83 | -19.54 (0.8) | 23.52 (0.4) |
| | $\beta$ | 25 | 0.84 | 0.36 | 1.00 | 1.51 | -28.39 (0.8) | 22.49 (0.5) |
| | $\beta$ | 20 | 0.86 | 0.07 | 1.06 | 0.66 | -45.62 (1.2) | 39.41 (0.7) |
| | $\beta$ | 15 | 0.11 | 0.00 | 0.24 | -0.01 | 32.62 (2.3) | 127.25 (1.1) |

**Table 2.** Intercomparison statistics for the time series in Figs. 6, 7, and 8. The subscripts PE and BG indicate measurements impacted by the Pinatubo eruption (June 1991-December 1997) and background conditions, respectively. Line of best fit was calculated with SAGE as the independent variable. Values in parentheses under the %Diff columns indicate standard error of the percent difference. NOTE: OHP has two products listed ($\beta$ and $S$ where $S$ uses the conventional definition), hence the two entries.

prior to the 1991 eruption and later in the record. In contrast to the lidar record, the SAGE record remained smooth throughout except at 30 km where it showed more variability.

It was observed that during the Pinatubo time period the coefficients of correlation and line-of-best fit slopes were higher than during background conditions. This was expected behavior for background conditions for two reasons: 1. in the absence of stratospheric injections the instruments were left to sample the natural stratospheric variability (similar to noise), which limits correlative analysis outside long-duration climatological trend studies, 2. the limited dynamic range of the observations provides, essentially, a correlation between two parallel lines. Overall, the percent differences for TMO show the two techniques to be in good agreement, with the worse agreement occurring at 15 km, which was expected due to cloud contamination.

### 4.1.2 Observatoire de Haute-Provence

Unlike TMO, the OHP lidar record did not start until $\approx$2.5 years into the Pinatubo recovery (similar to MLO). However, SAGE II recorded significantly more profiles over this latitude than over MLO, leading to a better representation of the zonal aerosol

loading throughout the month. The increased differences at 15 and 30 km was expected, as discussed above. However, we did not anticipate the large difference at 25 km when the atmosphere was impacted by Pinatubo (-29.44%). After further analysis it was determined that this difference was driven by a single, 2.5 year, time period that straddled both the Pinatubo time period and the beginning of the quiescent period (June 1996–January 1999). During this time the two records were consistently in substantial disagreement. This disagreement can be seen visually in Fig. 7. Removing data from this time period reduced the percent difference to -16.87% (percent difference during background conditions was reduced to +2.90%). In an attempt to identify the source of this discrepancy we repeated the analysis under different longitude criteria (e.g. instead of doing zonal means we used only SAGE profiles collected within 5, 10, and 20 degrees longitude), weekly means instead of monthly, and adjusted the temporal coincident criteria. The intention of this analysis was to determine whether variability that was local to OHP was driving the differences. However, we were unable to identify any such local variability and we currently cannot account for this anomaly within the time series.

In addition to $\beta$, the OHP data record contained a lidar ratio time series, thereby allowing comparison with the lidar ratio derived from the EBC algorithm. Percent differences for lidar ratio comparison are presented in Table 2. The slope and $R^2$ values were not reported for $S$ because the OHP $S$ value was held static for extended periods of time, making these statistics meaningless. However, the relative difference retains meaning, and we observe that the percent difference between $S$ values was consistently within 20%. Changes in the lidar ratio due to changing aerosol mode radii throughout the recovery time period were in agreement with what is expected due to a major volcanic eruption. Indeed, by the end of the SAGE II mission $S$ had recovered from the El Chichón and Pinatubo eruptions to a value of $\approx$50-60 at all altitudes as supported by the SAGE-derived $S$ and the estimate used in the lidar retrievals.

### 4.1.3  Mauna Loa

Similar to OHP, the MLO record did not begin until $\approx$2.5 years into the Pinatubo recovery. Beginning in June-1995 the two data sets began to diverge at 20 km (Fig. 8) with the lidar record flattening out . In contrast, $\beta_{SAGE}$ continued with a quasi-exponential decay until January 1998, in agreement with the other two sites as well as previously published studies (e.g. Deshler et al. (2003) and Thomason et al. (2018)). In January 1998 the lidar signal experienced an anomaly wherein the signal decreased by approximately an order of magnitude. After this time, $\beta_{SAGE}$ was consistently larger than $\beta_{Lidar}$. The discrepancy from June 1995-January 1998 at 20 km is currently not understood. However, the sudden change in January 1998 coincides with a new lidar instrument setup.

The statistics in Table 2 show the MLO comparison to be the worst of the three stations (excluding the -29.44% difference at 25 km over OHP; conversely the MLO percent difference at 25 km was relatively small). In addition to the anomalous behavior between 1995 and 1998 the SAGE II instrument experienced relatively few overpasses over Mauna Loa's latitude (19.5 N) as seen in Fig. 1 (A). Therefore, we suggest that the poor agreement between the two instruments may have been driven by inadequate sampling by SAGE.

## 4.2 SAGE III/ISS

To date, the SAGE III mission has made observations under relatively clean stratospheric conditions similar to conditions at the end of the SAGE II mission. Due to the limited data record (three years since launch), the comparison between SAGE III and the Mauna Loa and OHP lidars will be cursory. Data from the Table Mountain Facility have not been released for this time period, therefore Table Mountain was excluded from the current analysis.

The SAGE III and lidar backscatter coefficients show similar qualitative agreement at both Mauna Loa and OHP (Figs. 9 and 10, respectively) similar to what was observed in the SAGE II comparison (vide supra). During the SAGE III mission the atmosphere has been relatively stable, with a minor increase in backscatter/aerosol extinction in late 2017 due to a significant pyrocumulonimbus (pyroCB, indicated by vertical line in figures) event in northwestern Canada, which was comparable to a moderate volcanic eruption (Peterson et al., 2018). Smoke from the pyroCB was clearly visible over mid-latitude sites like OHP (Khaykin et al., 2017) while there was no clear evidence of significant aerosol loading at low latitudes (i.e. over Mauna Loa). However, Chouza et al. (2020) showed a small increase in stratospheric aerosol optical depth over Mauna Loa during this period.

Similar to the end of the SAGE II record, calculation of a meaningful $R^2$ value is likewise challenging when the variability is small. Further, getting good agreement between extreme-low $\beta$ values is challenging since small fluctuations have a disproportionate impact on the overall percent difference (see Table 3). However, this may be indicative of two possible conclusions: 1. the EBC method has limited applicability to background conditions, 2. the precision/accuracy of SAGE III or the ground-based lidar is too limited to make meaningful measurements during background conditions. The validity of option 1 can be challenged with the SAGE II record (compare especially Table 2) wherein it was shown that the background percent differences were generally better than during the Pinatubo period. Therefore, it would seem that the EBC method is applicable to quiescent periods. Considering the precision of the SAGE instruments and the number of lidar validation and intercomparison campaigns the possibility of option 2 being valid seems unlikely.

### 4.2.1 Overall Impression

For the SAGE II instrument the derived $\beta_{SAGE}$ products generally had high coefficients of correlation and slopes near 1 when compared with the lidar-derived products, especially in the 20-25 km range during background conditions. While agreement was consistently good in the 20 and 25 km bins (within 5%) the agreements consistently diverged in the 15 and 30 km bins for both Pinatubo and background time periods. The divergence at 15 km is likely attributable to optical depth and cloud contamination, but the divergence at 30 km is not as easily explained. Indeed, it may be partly caused by lack of return signal in the lidar and lack of optical depth for SAGE (though this is generally not a challenge for SAGE instruments at this altitude). We note that this divergence was modest ($\pm \approx 20\%$) over TMO and MLO, where $\beta$ was calculated using the BSR technique. However, the divergence was significantly larger over OHP where $\beta$ was calculated via the Fernald-Klett method and the lidar ratio was held constant and the atmosphere is considered aerosol free from 30-33 km. We suggest that this highlights the sensitivity of Fernald-Klett to atmospheric variability and the need to make an informed selection of lidar ratio.

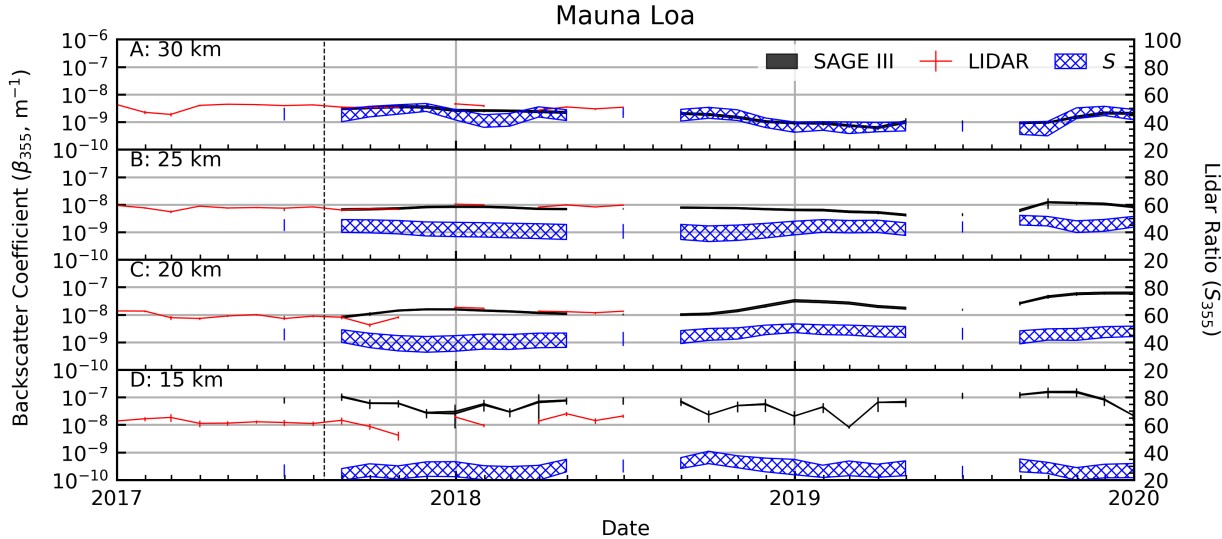

**Figure 9.** Same as Fig. 8, but for SAGE III. Vertical dashed line indicates the 2017 pyroCB event.

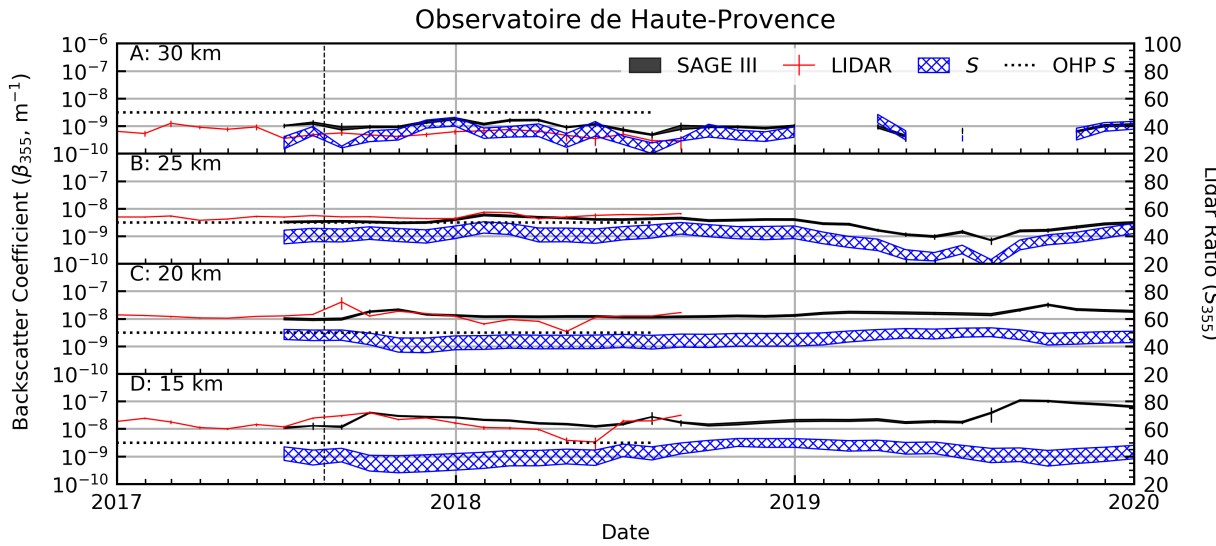

**Figure 10.** Same as Fig. 7, but for SAGE III. Vertical dashed line indicates the 2017 pyroCB event.

Perhaps the most striking feature of this analysis is how well the SAGE-derived backscatter coefficient agreed with the lidar record during the early stages of the Pinatubo eruption (Fig. 6) when particle shape and composition deviated significantly from our initial assumptions (Sheridan et al., 1992). This seems to indicate that using an extinction ratio may compensate for mischaracterization of size and composition assumptions within our model. However, further evaluation involving major

| Site | Parameter | Altitude (km) | $R^2$ | Slope | %Diff |
|------|-----------|---------------|-------|-------|-------|
| OHP | $\beta$ | 30 | 0.45 | 0.25 | 74.27 (1.6) |
|     | $\beta$ | 25 | 0.46 | 0.80 | -29.93 (1.1) |
|     | $\beta$ | 20 | 0.00 | -0.14 | 3.03 (3.5) |
|     | $\beta$ | 15 | 0.22 | 0.60 | 18.02 (4.1) |
| OHP | $S$ | 30 | — | — | -37.09 (1.1) |
|     | $S$ | 25 | — | — | -3.17 (0.3) |
|     | $S$ | 20 | — | — | -8.86 (0.4) |
|     | $S$ | 15 | — | — | -19.86 (0.5) |
| MLO | $\beta$ | 30 | 0.00 | 0.04 | -26.80 (2.3) |
|     | $\beta$ | 25 | 0.48 | 1.53 | -14.19 (1.6) |
|     | $\beta$ | 20 | 0.39 | 1.16 | 6.65 (4.3) |
|     | $\beta$ | 15 | 0.01 | 0.03 | 121.53 (4.9) |

**Table 3.** Same as Table 2, but for SAGE III. All conditions were classified as background.

volcanic eruptions is required to better understand whether this agreement is fortuitous or the EBC algorithm is actually insensitive to aerosol composition and shape.

The calculated $S$ at each site was in good agreement with values calculated by Jäger et al. (1995). Immediately prior to the eruption $S$ was approximately 40-45 for the lowermost altitudes (tropopause-20 km) and slightly higher (50-60) in the 25-30 km altitudes. This was followed by a quick decrease after the eruption of Pinatubo, down to values of 20 in the Jäger dataset, with our calculated value being slightly lower. Overall, the calculated $S$ shows good agreement with the Jäger dataset in both magnitude and trend with altitude. Other studies that did not overlap with either SAGE II or SAGE III have shown similar $S$ values to those calculated here (Bingen et al., 2017; Painemal et al., 2019).

## 5 Conclusions

A method of converting SAGE extinction ratios to backscatter coefficient ($\beta$) profiles was presented. The method invoked Mie theory as the conduit from extinction to backscatter space and required assumptions on particle shape (spherical), composition (75% water, 25% sulfuric acid), and distribution shape (single-mode log-normal with distribution width ($\sigma$) of 1.5). The general behavior of the model as a function of $\sigma$ was briefly considered (Fig. 2 and §3). It was demonstrated that, due to improper selection of $\sigma$, the corresponding $\beta$ value could be off by up to +32/-16% when the extinction ratio exceeds 6, but that >90% of the SAGE II and SAGE III records had extinction ratios <6.

A major finding of this research was the demonstration of the robustness of the conversion method. It was shown that, within the specified error bars, the calculation of $\beta$ was independent of SAGE wavelength combination (§3.1). Further, we showed that

when altitude was limited to 15-31 km the robustness improved significantly (Table 1). Therefore, we recommend limiting the use of this conversion method to this altitude range unless appropriate modifications can be made to improve the consistency of its performance at higher/lower altitudes. Such improvements may include cloud screening at low altitudes and appropriate adjustment of size distribution parameters at higher altitudes.

The robustness of the conversion method provides an indirect validation of the SAGE aerosol spectra. If the EBC method were wavelength dependent, this would indicate a substantial error in the standard aerosol products. However, our evaluation showed that the EBC is not wavelength dependent, thereby lending credence to the SAGE aerosol products wavelength assignment.

It was shown that, overall, the SAGE II-derived $\beta$ product was in good agreement with the lidar data during both background (percent difference within $\approx$10%) and elevated (percent difference within $\approx$10-20% depending on location). Indeed, we showed that this agreement was altitude dependent, with better agreement in the middle stratosphere and worse agreement at lower (15 km) and upper (30 km) altitudes. The reason for this divergence was attributed to increased optical depth and cloud contamination at low altitudes and decreased aerosol load at higher altitudes. The lack of optical depth at high altitudes is less of a problem for SAGE than for the lidar. This is due, fundamentally, to the differing viewing geometries: SAGE retains a long observation path length at 30 km, while the lidar instrument relies on few photons being backscattered at 30 km. Further, all scattered photons must re-pass through the most dense portion of atmosphere (without being absorbed or scattered) prior to impinging on the lidar detector. This limitation is most readily observed by considering how the noise/variance increases with altitude in a lidar profile.

For the SAGE III analysis only OHP and MLO were available for comparison. The SAGE III-derived $\beta$ product showed worse agreement than the SAGE II data. The lower $R^2$ values were attributed to lack of variability within the data records (e.g. $R^2$ of parallel straight lines is 0). However, the larger percent differences may have been due to the magnitude of the backscatter values (e.g. small differences (e.g. 2E-10) for small numbers (e.g. 1E-10) yield large percent differences (here, 100%)). Another consideration is that the SAGE III record, to date, is short compared to SAGE II and the lidar coverage within the SAGE III time period is approximately one year, further limiting the intercomparison. As the record expands (possibly including observation of moderate-to-major volcanic events) we expect the comparison with the lidar data to improve.

A potential application of this method is in informing lidar ratio ($S$) selection for lidar observations. As an example, processing for the Cloud-Aerosol Lidar and Infrared Pathfinder Satellite Observation (CALIPSO) lidar currently assumes a static lidar ratio (50 sr) for all latitudes and all altitudes. As was recently shown by Kar et al. (2019), CALIPSO extinction products have an altitudinal and latitudinal dependence as compared to SAGE III. Providing better $S$ values may improve this agreement and may be beneficial in processing CALIPSO $\beta$ products as well.

Another application of this method may be in providing global backscatter profiles independent of a space-based lidar such as CALIPSO. While we do not suggest that SAGE-derived backscatter coefficients can replace lidar observations, our product may be a viable alternative. With CALIPSO scheduled for decommissioning no later than 2023 (M. Vaughan, personal communication, 2020), and no replacement scheduled for flight prior to its decommissioning date, the SAGE III backscatter product may provide a necessary link between CALIPSO and the next space-based lidar to ensure continuity of record and

provide a method of evaluating the performance of the next-generation orbiting lidar in the context of the SAGE III record and, by association, CALISPO.

*Data availability.* SAGE data used within this study are available on NASA's Atmospheric Science Data Center (https://eosweb.larc.nasa.gov/).
450 The lidar data used in this study are available from the NDACC archive (https://www.ndacc.org).

*Author contributions.* TNK and LT developed the methodology, while TNK carried out the analysis, wrote the analysis code and the manuscript. TL and FC provided lidar data collected at TMF and MLO and assisted in the description of this data product in the manuscript. SK and SGB provided lidar data collected over OHP and assisted in the description of this data product in the manuscript. MR, RD, KL, and DF participated in scientific discussions and provided guidance throughout the study. All authors reviewed the manuscript during the
455 preparation process

*Competing interests.* The authors declare that they have no competing interests

*Acknowledgements.* SAGE III/ISS is a NASA Langley managed mission funded by the NASA Science Mission Directorate within the Earth Systematic Mission Program. Enabling partners are the NASA Human Exploration and Operations Mission Directorate, International Space Station Program and the European Space Agency. SSAI personnel are supported through the STARSS III contract NNL16AA05C.

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
