# Peer review of "Evaluation of a Method for Converting SAGE Extinction Coefficients to Backscatter Coefficient for Intercomparison with LIDAR Observations"

_Atmospheric Measurement Techniques, 2020_

## Short Comment (SC1) · 19 Mar 2020

Estimating the PSD from Mie calculation based on assumptions of composition (75%-25%), particle shape (spherical) and size distribution (lognormal), based on a lookup table as explained in ll. 124-133 is a method that has been developed for the case of SAGE II by Bingen et al. (2004). These previous studies are thus an obvious precursor of the present work and should be duly cited.

Furthermore, it is surprising that the authors claim that the SAGE derived backscatter coefficient "will be independent of wavelength combination", since "it can be trivially demonstrated that, working strictly within the confines of theory, this is the case" (ll

166-167). What the authors mean here is not clear. On the contrary, it is known that Mie theory is valid for spherical particles with a size in the same order of magnitude as the wavelength. In this respect, using wavelengths of 385 of 1020 nm (as in Eq. (4)) is not equivalent at all, the extinction coefficient at the different wavelengths being particularly sensitive to different size ranges (Bingen et al., 2002), and possibly, to different modes present in the aerosol population. Hence, using a wavelength as close as possible to the lidar wavelength (355 nm) in Eq. (4) should be the best choice to provide a coherent conversion of the extinction measurements to an estimate of the backscatter coefficient.

Also, the authors do not validate of their estimate of the lidar ratio, although several studies provide comparisons data. For instance, Vernier et al. (2011) derived a climatology of extinction-to-backscatter ratio based on GOMOS and CALIPSO measurements at 525 nm. Also, Bingen et al. (2017) present an intercomparison between GOMOS aerosol extinction coefficient and lidar measurements from several ground-based stations including Mauna Loa, and discuss the results of these intercomparisons as a function of the choice of extinction-to-backscatter ratio. Finally, Painemal et al. (2019) published lidar ratios above oceans retrieved from CALIPSO and CloudSat. These results would be usefully compared to the results of the present study to assess the robustness of lidar ratio estimates from these satellite measurements.

Bingen, C., F. Vanhellemont, and D. Fussen, A new regularized inversion method for the retrieval of stratospheric aerosol size distributions applied to 16 year SAGE II data (1984–2000): Method, results and validation, Ann. Geophys., 21, 797– 804, 2002.

Bingen, C., D. Fussen, and F. Vanhellemont, A global climatology of stratospheric aerosol size distribution parameters derived from SAGE II data over the period 1984–2000: 2. Reference data, J. Geophys. Res., 109, D06202, doi:10.1029/2003JD003511, 2004.

Bingen, C., Robert, C. E., Stebel, K., Brühl, C., Schallock, J., Vanhellemont, F.,

Mateshvili, N., Höpfner, M., Trickl, T., Barnes, J. E., Jumelet, J., Vernier, J.-P., Popp, T., de Leeuw, G., and Pinnock, S.: Stratospheric aerosol data records for the climate change initiative: Development, validation and application to chemistry-climate modelling, Remote Sens. Environ., 203, 296– 321, 2017.

Painemal,D., M. Clayton, R. Ferrare, S. Burton, D. Josset, and M. Vaughan, Novel aerosol extinction coefficients and lidar ratios over the ocean from CALIPSO–CloudSat: evaluation and global statistics, Atmos. Meas. Tech., 12, 2201-2217, doi:10.5194/amt-12-2201-2019, 2019

Vernier, J.-P., T.D. Fairlie, M. Natarajan, F.G. Wienhold, J. Bian, Increase in upper tropospheric and lower stratospheric aerosol levels and its potential connection with Asian pollution, J. Geophys. Res. Atmos., p. 120, doi:10.1002/2014JD022372, 2015

---

## Referee Comment (RC1) · Anonymous Referee #1 · 7 Apr 2020

This is a well written paper dealing with a very important issue in observing strato-spheric aerosols. The paper analyzes the SAGE II and SAGE III ISS extinction data at multiple wavelengths, makes very reasonable assumptions for a Mie scatter model, and then computes the 180 degree backscatter. SAGE II is a very long and important data set and hopefully if SAGE III operates for many years it will be equally valuable. Lidars can complement the satellite aerosol measurements by giving measurements at a single location with high time and altitude resolution. But the valuable quantity is extinction, not the lidar 180 degree backscatter, so the Lidar Ratio (Ext/Backscatter) is needed for the conversion. The paper analyzes the SAGE data to show different wavelengths give self-consistent results, and quantifies the errors. The time period

includes both the Mt. Pinatubo eruption and the relaxation to background conditions. The Lidar Ratio is seen to smoothly change during the eruption which is very useful for interpreting lidar data taken during the eruption. The only major comment is that the backscatter wavelength from the three lidars is 355 nm. Lidar measurements at 532 nm may be even more numerous and have an advantage in signal to noise. The fundamental lidar quantity is the ratio (molecular + aerosol)/molecular scatter. The aerosol scatter at 355 and 532 nm will be similar but the molecular scatter at 355 will be 5 times stronger than at 532 nm. So you are effectively looking at smaller changes in the ratio. It would be valuable to repeat the analysis for 532 nm backscatter.

Interactive comment by Bingen: If I understand her correctly Bingen is saying this paper implies beta is not wavelength dependent, which it clearly is. Line 166 states that beta(Sage) is independent of wavelength combination. I take this to mean that different combinations of the SAGE wavelengths can be used in the analysis and the results are the same with error bars. This is a strong result of the paper and maybe it can be explained better.

In the second part of Bingen's comment she makes the point that a comparison of S should be made with this paper and previous papers. I would agree. Papers by Jaeger et al. (already referenced), Bingen (referenced in comment), and Altuna (below) are ones I know of.

Antuña, J. C., A. Robock, G. Stenchikov, J. Zhou, C. David, J. Barnes, and L. Thomason, Spatial and temporal variability of the stratospheric aerosol cloud produced by the 1991 Mount Pinatubo eruption, J. Geophys. Res., 108(D20), 4624 (2003).

Antuña, J. C., A. Robock, G. L. Stenchikov, L. W. Thomason, and J. E. Barnes, Lidar validation of SAGE II aerosol measurements after the 1991 Mount Pinatubo eruption, J. Geophys. Res., 107(D14), 4194 (2002).

Specific comments:

Line 72: charge coupled device (with) a resolution ... Line 88: typo ...Backscatter ...
Line 128: Why was this sigma range chosen? Is there a reference? Line 141: You
might add a note that although Ext/beta is defined as S, Ext and beta are normally at
the same wavelength. In this paper you sometimes use this quantity with two different
wavelengths. Figure 6, 7, & 8: The legend overwrites some of the data. Can this be
fixed?

---

## Referee Comment (RC2) · Anonymous Referee #2 · 8 Apr 2020

Review for "Evaluation of a Method for Converting SAGE Extinction Coefficients to Backscatter Coefficient for Intercomparison with LIDAR Observations" by T. Knepp et al.

The authors present a calculation of Aerosol backscatter coefficients using multi-wavelength aerosol extinction products from the SAGE II and SAGE III/ISS instruments. The conversion methodology is presented followed by an evaluation of the conversion algorithm's robustness. I've tried to offer suggestions below that will improve methodology and overall clarity of the manuscript.

General comments: P1L14: Write out abbreviations for the first case use (e.g. SO2

(sulfur dioxide)).

P2L36: The discussion on Lidar Ratio, S, needs improvement and more adequate literature referencing. For instance, there is no discussion of spectral differences between SAGE/lidars and how that may impact the lidar ratio. The value of S greatly depends on the wavelength of laser used. There is also no description on how you will interpolate/extrapolate an altitude varying S.

P4L80: I agree this effort is likely more important to do with the three selected ground based lidars. However, it woud be useful to have a follow on study utilizing additional data sets from NASA's MPLNet or the European EARLINet. An ideal case (perhaps there is a case study already) would be to use an event where two lidar sites could provide the vertical distribution of aerosols coincident with a SAGE occultation observation.

P6L125: How can you parse out the vertical distribution of aerosols vs. the horizontal inhomogeneity? There needs to at least be a description of the uncertainty associated with the measurement EBC assumptions. Will water vapor contamination in the longer wavelength bands become a source of further uncertainty?

I'm hesitant that a single value will be able to account for these. For a paper that leans so heavily on the assumption of sphericity in particles, I was surprised to not see a single mention of aerosol polarization/depolarization measurements from either ground-based of spaceborne (CALIPSO, CATS) instruments. These have long been known to provide context for optical and microphysical properties of aerosols. This manuscript could benefit from a short case study in which the authors show a proof of concept with a known event, rather than just grab bulk aggregate statistics that have no physical meaning.

There is recent work describing non-sphericity of Volcanic ash in the stratosphere. In particular, Noh et al., 2017 describe the settling of non-spherical particles after volcanic eruptions as well as a time dependence on the sphericity since eruption.

See (among others):

Michael I. Mishchenko, Janna M. Dlugach, Andrew A. Lacis, Larry D. Travis, and Brian Cairns, "Retrieval of volcanic and man-made stratospheric aerosols from orbital polarimetric measurements," Opt. Express 27, A158-A170 (2019)

Noh, Young Min, Dong Ho Shin, and Detlef Müller. "Variation of the vertical distribution of Nabro volcano aerosol layers in the stratosphere observed by LIDAR." Atmospheric environment 154 (2017): 1-8

Figure 3: Does this suggest that there are geophysical differences in the aerosol loading during the SAGE II/III time periods? Should the width be the same in A/C in non-Pinatubo times?

Figure 6: Is the SAGE data noisier towards the end of the record? Are results any different if you remove the last year? In general, is there something geophysical occurring that is decreasing the spread of the S value over time or that simply lack of signal? Also, the legend is obscuring the data.

Figure 8: Is there an explanation for the discrepancy in MLO BC after ∼2000? Is this cloud contamination? It looks to be consistent – was there some calibration that was changed? The S value at 15km at MLO is much lower after 2000 than other site, which does not seem reasonable for stratospheric aerosol which would largely be well mixed. I'm sure this was verified but are the altitude layers for SAGE and lidar both in ASL?

---

## Referee Comment (RC3) · Anonymous Referee #3 · 8 May 2020

General comments on Knepp et al. [2020]:

This paper presents a novel way to compare the "native" products of two types of stratospheric aerosol data: The extinction coefficients derived from SAGE measurements and the backscattering coefficients derived from LIDAR measurements. The method is carefully evaluated based on the statistical properties of the SAGE data product, and the paper provides clear guidance for the reader who is interested in applying the method.

Detailed comments:

Line 24: "This technique allows for high-precision measurements on the order of 5%

for aerosol extinction. . ." – This statement requires attribution.

Line 123: Assuming that the PSD is single-mode log-normal definitely reduces the problem to a manageable solution space, but was any analysis done to estimate how your conclusions might change if you assumed some other model? In particular, in-situ stratospheric aerosol observations (see http://www.atmosp.physics.utoronto.ca/SPARC/index.html, for example) frequently show multiple modes, and the different particle sizes represented in those modes clearly have the potential to affect the extinction and phase function differently.

Line 127: The symbol r_m is frequently called the "mode radius" in the aerosol literature, but it actually represents the median of the distribution (See Johnson, Norman L.; Kotz, Samuel; Balakrishnan, N. (1994), "14: Lognormal Distributions", Continuous univariate distributions. Vol. 1, Wiley Series in Probability and Mathematical Statistics: Applied Probability and Statistics (2nd ed.), New York: John Wiley Sons.)

Line 129: ". . . a new log-normal distribution as r_m took on each value within r." Is this accurate? If so, then you considered distributions for which r_m occurred at the smallest (and largest) possible particle size? This seems ill-advised from a mathematical perspective (& doesn't really yield a "log-normal" distribution in any meaningful sense). Your solutions lie comfortably in the middle of the range given, so this is probably a minor point, but perhaps the description should be re-written?

Line 159: Were any comparisons made between the derived beta_355 and the SAGE measurement of aerosol extinction at 385 nm? That SAGE product is generally understood to be lower in quality than the aerosol extinction at longer wavelengths, but it is a bit strange not to use it, or even mention its existence as an option.

Figure 2 is a bit blurry.

Figure 3: I'm not sure that I understand the meaning of the dark red vs. light red vs. gray regions.

[Figure]

Line 280 – This "atmospheric opacity and cloud contamination" concerns here clearly correspond to SAGE measurements, correct?

Line 323 – Is there a reference for the details of the "new lidar instrument setup" mentioned here? It might provide helpful context for the sudden change observed in the comparison.

Line 337 – "Smoke from the pyroCB was visible over OHP, but not over Mauna Loa." This seems reasonable, but what evidence is it based on?

Line 342 – "precision/accuracy . . . is too limited to make meaningful measurements during background conditions." This is an alarming statement on its face, but I assume you only mean to exclude the possibility of meaningful measurements of the beta_355 parameter under discussion.

Technical comments:

General - The units of the lidar ratio (S) should be presented consistently (sometimes it has no units, sometimes sr).

Line 88 – "Bakckscatter"

Line 278 / Equation 5 – Lidar should be subscripted in denominator

---

## Author Comment (AC1) · 29 May 2020

We would like to thank Dr. Bingen for providing feedback on this manuscript. Our responses are provided below (red) to her suggestions (black).

Estimating the PSD from Mie calculation based on assumptions of composition (75%-25%), particle shape (spherical) and size distribution (lognormal), based on a lookup table as explained in ll. 124-133 is a method that has been developed for the case of SAGE II by Bingen et al. (2004). These previous studies are thus an obvious precursor

[Figure]

of the present work and should be duly cited.

Bingen et al. 2004 is not a direct precursor to this study since they used SAGE extinctions and Mie theory to derive particle size distribution (PSD) parameters, while we use Mie theory to form a bridge between extinction and backscatter without attempting to retrieve PSD parameters. Therefore, we do not believe citing Bingen et al. 2004 is appropriate at this specific point in the text. However, we have updated the introduction to add clarification, which includes adding Bingen et al. 2004 as a reference.

Furthermore, it is surprising that the authors claim that the SAGE derived backscatter coefficient "will be independent of wavelength combination", since "it can be trivially demonstrated that, working strictly within the confines of theory, this is the case" (ll 166-167). What the authors mean here is not clear. On the contrary, it is known that Mie theory is valid for spherical particles with a size in the same order of magnitude as the wavelength. In this respect, using wavelengths of 385 of 1020 nm (as in Eq. (4)) is not equivalent at all, the extinction coefficient at the different wavelengths being particularly sensitive to different size ranges (Bingen et al., 2002), and possibly, to different modes present in the aerosol population. Hence, using a wavelength as close as possible to the lidar wavelength (355 nm) in Eq. (4) should be the best choice to provide a coherent conversion of the extinction measurements to an estimate of the backscatter coefficient.

This statement was perhaps poorly worded. We have updated to the text for clarification. What we intended to claim was that under ideal conditions the selection of extinction wavelength combination would have no impact on the derived backscatter coefficient (within the error limits) and that when dealing with theoretical data (which has no errors or uncertainties) this is easily demonstrated. However, we go on to state that the SAGE dataset does have error sources, meaning wavelength selection *might* be important. We devoted the entirety of §3.1 to evaluating the impact of wavelength
selection on the calculated backscatter.

Also, the authors do not validate of their estimate of the lidar ratio, although several studies provide comparisons data. For instance, Vernier et al. (2011) derived a climatology of extinction-to-backscatter ratio based on GOMOS and CALIPSO measurements at 525 nm. Also, Bingen et al. (2017) present an intercomparison between GOMOS aerosol extinction coefficient and lidar measurements from several ground-based stations including Mauna Loa, and discuss the results of these intercomparisons as a function of the choice of extinction-to-backscatter ratio. Finally, Painemal et al. (2019) published lidar ratios above oceans retrieved from CALIPSO and CloudSat. These results would be usefully compared to the results of the present study to assess the robustness of lidar ratio estimates from these satellite measurements.

The discussion regarding lidar ratio agreement to other measurements has been updated and the reference list augmented.

---

## Author Comment (AC2) · 29 May 2020

We thank the reviewer for reading this manuscript and providing feedback. Below are our responses to the reviewer's comments. Reviewer's comments are in black, our responses are in red.

The only major comment is that the backscatter wavelength from the three lidars is 355 nm. Lidar measurements at 532 nm may be even more numerous and have an advantage in signal to noise. The funda- mental lidar quantity is the ratio (molecular

[Figure]

+ aerosol)/molecular scatter. The aerosol scatter at 355 and 532 nm will be similar but the molecular scatter at 355 will be 5 times stronger than at 532 nm. So you are effectively looking at smaller changes in the ratio. It would be valuable to repeat the analysis for 532 nm backscatter

We agree that using backscatter coefficients at 532 nm would make an interesting follow-on study as suggested by both anonymous reviewers. Indeed, application of this technique to 532 nm lidar products will be the subject of a subsequent publication.

Interactive comment by Bingen: If I understand her correctly Bingen is saying this paper implies beta is not wavelength dependent, which it clearly is. Line 166 states that beta(Sage) is independent of wavelength combination. I take this to mean that different combinations of the SAGE wavelengths can be used in the analysis and the results are the same with error bars. This is a strong result of the paper and maybe it can be explained better.

Your understanding is correct. This is a major finding of the manuscript and has the entirety of section 3.1 dedicated to this. The wording at the end of §3.1 and in the conclusions was updated to more clearly communicate this.

In the second part of Bingens comment she makes the point that a comparison of S should be made with this paper and previous papers. I would agree. Papers by Jaeger et al. (already referenced), Bingen (referenced in comment), and Altuna (below) are ones I know of.

Additional text and references have been added to discuss the lidar ratio agreement.

charge coupled device (with) a resolution Corrected

Line 88: typo . . . Backscatter Corrected

Line 128: Why was this sigma range chosen? Is there a reference?

The range of sigmas was chosen to be representative of the reasonable range expected in the stratosphere. This is now explicitly stated in the text with references.

Line 141: You might add a note that although Ext/beta is defined as S, Ext and beta are normally at the same wavelength. In this paper you sometimes use this quantity with two different wavelengths.

Clarification was added to §3 and figure/table captions.

Figure 6, 7, & 8: The legend overwrites some of the data. Can this be fixed? Corrected

---

## Author Comment (AC3) · 29 May 2020

We thank the reviewer for reading this manuscript and providing feedback. Below are our responses to the reviewer's comments. Reviewer's comments are in black, our responses are in red.

P1L14: Write out abbreviations for the first case use (e.g. SO2)

Corrected

[Figure]

P2L36: The discussion on Lidar Ratio, S, needs improvement and more adequate literature referencing. For instance, there is no discussion of spectral differences between SAGE/lidars and how that may impact the lidar ratio. The value of S greatly depends on the wavelength of laser used. There is also no description on how you will interpolate/extrapolate an altitude varying S.

This paragraph has been updated to point out spectral differences between SAGE and lidar and the references have been updated as well.

P4L80: I agree this effort is likely more important to do with the three selected ground based lidars. However, it woud be useful to have a follow on study utilizing additional data sets from NASA's MPLNet or the European EARLINet. An ideal case (perhaps there is a case study already) would be to use an event where two lidar sites could provide the vertical distribution of aerosols coincident with a SAGE occultation observation.

We agree that a follow up study using MPLNet or EARLIENet or another lidar network would be interesting. However, we note the challenge in using MPLNet and EARLINet data as, from our evaluation of the data from these networks, data collection typically stops just beyond the tropopause. However, this method is being applied to ongoing research which will be the topic of future publications.

P6L125: How can you parse out the vertical distribution of aerosols vs. the horizontal inhomogeneity? There needs to at least be a description of the uncertainty associated with the measurement EBC assumptions. Will water vapor contamination in the longer wavelength bands become a source of further uncertainty?

The reference you provided (P6L125) does not correspond to anything related to this

comment. However, this comment is relevant and has been addressed in the updated manuscript. First, the uncertainty associated with the EBC assumptions were addressed in section 3.2 of the original manuscript. Second, to date, similar studies that compared SAGE profiles with sonde or lidar data have taken advantage of the horizontal homogeneity of the atmosphere. This has been explained, with appropriate supporting references, in the updated version of the manuscript. Regarding water vapor, this is not a factor since water vapor's absorption at 1020 nm is insignificant compared to aerosol extinction as stated in Damadeo et al. 2013.

I'm hesitant that a single value will be able to account for these. For a paper that leans so heavily on the assumption of sphericity in particles, I was surprised to not see a single mention of aerosol polarization/depolarization measurements from either ground-based of spaceborne (CALIPSO, CATS) instruments. These have long been known to provide context for optical and microphysical properties of aerosols. This manuscript could benefit from a short case study in which the authors show a proof of concept with a known event, rather than just grab bulk aggregate statistics that have no physical meaning.

We agree that a using both CATS and CALIOP for an extended/detailed examination of this methodology would be interesting and scientifically beneficial. Further, data from the CALIPSO lidar would provide valuable information and an interesting series of case studies. The challenge in using CALIPSO lidar data is the poor signal-to-noise ratio for aerosol backscatter in the stratosphere. This is currently being investigated and will be the subject of a future publication.
The CATS lidar may provide another dataset for evaluation. However, achieving appropriate coincidence is challenging due to both instruments (SAGE III and CATS) being mounted to the ISS.

Figure 3: Does this suggest that there are geophysical differences in the aerosol loading during the SAGE II/III time periods? Should the width be the same in A/C in non-Pinatubo times?

Yes, this indicates a geophysical difference. This is due to differing levels of volcanic activity. Additional comments have been added to the text.

Figure 6: Is the SAGE data noisier towards the end of the record? Are results any different if you remove the last year? In general, is there something geophysical occurring that is decreasing the spread of the S value over time or that simply lack of signal? Also, the legend is obscuring the data.

The last year has no impact on the analysis by itself. The spread in lidar ratios decreased because of changes in extinction coefficients. This was caused for two reasons: 1. the variability of extinction observations was higher early in the record (product of El Chichon and subsequent eruptions) and decreased as quiescent conditions were achieved late in the SAGE II record; 2. as aerosol size distributions changed so too did the extinction ratio. The legend issue was corrected.

Figure 8: Is there an explanation for the discrepancy in MLO BC after âĹij2000? Is this cloud contamination? It looks to be consistent – was there some calibration that was changed? The S value at 15km at MLO is much lower after 2000 than other site, which does not seem reasonable for stratospheric aerosol which would largely be well mixed. I'm sure this was verified but are the altitude layers for SAGE and lidar both in ASL?

We see no discrepancy between panels B and C in Figure 8, so we are unsure of how to address this concern. We can say that some of the difference between MLO and other sites will be due to latitude (MLO being tropical, hence a tropopause on the order

of 15 km). Yes, all data sets had altitudes in ASL. From what we see everything in this figure is reasonable.

---

## Author Comment (AC4) · 29 May 2020

We thank the reviewer for reading this manuscript and providing feedback. Below are our responses to the reviewer's comments. Reviewer's comments are in black, our responses are in red.

Line 24: "This technique allows for high-precision measurements on the order of 5% for aerosol extinction..." - This statement requires attribution.
There is no single publication to reference this statement. However, the precision is reported in the data product. The text has been updated to reflect this.

Line 123:Assuming that the PSD is single-mode log-normal definitely reduces the problem to a manageable solution space, but was any analysis done to estimate how your conclusions might change if you assumed someother model? In particular, in-situ stratospheric aerosol observations (seehttp://www.atmosp.physics.utoronto.ca/SPARC/index.html, for example) frequentlyshow multiple modes, and the different particle sizes represented in those modes clearly have the potential to affect the extinction and phase function differently.

Particle size distributions (PSDs) with more than one mode were not evaluated within this work. We agree that using single-mode distributions greatly simplifies/reduces the solution space. The intent of this work was to evaluate how well derived backscatter coefficients agree with lidar products within the confines of our assumptions, one of which was single-mode distributions. We agree that most stratospheric, non-background, PSDs will have more than one mode; this was certainly the case after Pinatubo. Further, recent work by von Savigny & Hoffmann (AMT, 2020) may be indicative of backscatter being less sensitive to multi-modal PSDs than extinction. However, this certainly merits further analysis to reach a non-speculative conclusion.

Line 127: The symbol r_m is frequently called the "mode radius" in the aerosol literature, but it actually represents the median of the distribution (See Johnson, NormanL.; Kotz, Samuel; Balakrishnan, N. (1994), "14: Lognormal Distributions", Continuousunivariate distributions. Vol. 1, Wiley Series in Probability and Mathematical Statistics:Applied Probability and Statistics (2nd ed.), New York: John Wiley Sons.)

This is a great semantics point. I (TNK) have found the reference to "mode radius" in the literature to be confusing because, as the reviewer points out, this actually refers to median radius. We refer to r_m as mode radius to be consistent with the literature. However, clarification has been added to the text to aid the reader.

Line 129: "...a new log-normal distribution as r_m took on each value within r." Is this accurate? If so, then you considered distributions for which r_m occurred at the smallest (and largest) possible particle size? This seems ill-advised from a mathematical perspective (& doesn't really yield a "log-normal" distribution in any meaningful sense). Your solutions lie comfortably in the middle of the range given, so this is probably a minor point, but perhaps the description should be re-written?

There are two points that need addressed, neither of which impact the results of this work in any way. First, the radius range stated in the paper was incorrect. The range used to generate all tables, figures, etc. extended from 1 to 1500 nm. This was just a typing error in the manuscript (not the analysis code) and has been corrected. Second, the mode radii that were actually *needed* for this analysis (i.e. to regenerate the SAGE-based extinction ratios) were in the range of 50-500 nm, well away from these bounds. This has been clarified in the text.

Line 159: Were any comparisons made between the derived beta_355 and the SAGE measurement of aerosol extinction at 385 nm? That SAGE product is generally understood to be lower in quality than the aerosol extinction at longer wavelengths, but it isa bit strange not to use it, or even mention its existence as an option.

We are unsure of what the reviewer is asking here. The 385 channel was used in the analysis (used 385/1020 extinction ratio to estimate backscatter at 355). However, if

the reviewer is asking about a *direct* comparison of extinction at 385 to backscatter at 355, then no, we did not perform this comparison. This type of comparison is outside the scope of this manuscript since we were evaluating the performance of an extinction-to-backscatter algorithm.

Figure 2 is a bit blurry

The figure supplied to Copernicus is the same resolution as all other figures, so it should be of sufficient quality (it looks good on my copy as well). However, I will pay attention to this during the final printing to ensure sufficient quality and provide a higher-resolution image if needed. We appreciate the reviewer pointing this out since blurry figures can make interpretation challenging.

Figure 3: I'm not sure that I understand the meaning of the dark red vs. light red vs.gray regions.

The confusion may be caused by the legend only having two colors (light-red and gray). The dark-red region is where the two distributions overlap. The caption has been updated to clarify this.

Line 280 - This "atmospheric opacity and cloud contamination" concerns here clearly-correspond to SAGE measurements, correct?

Correct.

Line 323 - Is there a reference for the details of the "new lidar instrument setup" mentioned here? It might provide helpful context for the sudden change observed in the comparison.

The 1995-1998 period is an early period for the Mauna Loa system. Although technical problems during this period were reported in the public NDACC meta data archive, many minor setup changes were made without being reported in a publication. Therefore there is no specific reference we can provide besides the main [McDermid et al., 1995] publication pertaining to the original setup (we added this publication to the references). These things said, the sudden changes at the lower end of the profiles can only be caused by either slight beam/telescope misalignment, or spectral leakage in the 355 nm or 387 nm filters, or a combination of both.

Line 337 - "Smoke from the pyroCB was visible over OHP, but not over Mauna Loa."This seems reasonable, but what evidence is it based on?

A citation has been added regarding detection of smoke over OHP. Based on global observations (e.g., GloSSAC, OMPS/LP), there is no clear evidence of significant aerosol loading increase at low latitudes associated with the 2017 PyroCb plume, although Chouza et al. (2020) (we added this publication to the references) showed a very slight increase of sAOD values during that period.

Line 342 - "precision/accuracy...is too limited to make meaningful measurementsduring background conditions." This is an alarming statement on its face, but I assumeyou only mean to exclude the possibility of meaningful measurements of the beta_355 parameter under discussion.

Correct.

General - The units of the lidar ratio (S) should be presented consistently (sometimesit has no units, sometimes sr).

Updated throughout text.

Line 88 – "Bakckscatter"

Corrected

Line 278 / Equation 5 – Lidar should be subscripted in denominator

Corrected